# Effect of Enriched Environment on Cerebellum and Social Behavior of Valproic Zebrafish

Bernardo Flores-Prieto [1], Flower Caycho-Salazar [1], Jorge Manzo [2], María Elena Hernández-Aguilar [2], Alfonso Genaro Coria-Avila [2], Deissy Herrera-Covarrubias [2], Fausto Rojas-Dúran [2], Gonzalo Emiliano Aranda-Abreu [2], Cesar Antonio Pérez-Estudillo [2] and María Rebeca Toledo-Cárdenas [2,*]

[1] Doctorate in Brain Research, Universidad Veracruzana, Veracruz 91190, Mexico; bersebaflo@hotmail.com (B.F.-P.)
[2] Institute of Brain Research, Universidad Veracruzana, Veracruz 91190, Mexico
* Correspondence: rtoledo@uv.mx

**Abstract:** The etiology of autism spectrum disorder (ASD) has been linked to both genetic and epigenetic factors. Among the epigenetic factors, exposure to valproic acid (VPA), an antiepileptic and mood-modulating drug, has been shown to induce characteristic traits of ASD when exposed to during embryogenesis. Conversely, in animal models, enriched environment (EE) has demonstrated positive behavioral and neural effects, suggesting its potential as a complementary treatment to pharmacological approaches in central nervous system disorders. In this study, we utilized zebrafish to model ASD characteristics induced by VPA and hypothesized that sensory stimulation through EE could ameliorate the behavioral and neuroanatomical features associated with ASD. To test this hypothesis, we assessed social behavior, cerebellar volume, and Purkinje cell populations via histology and immunohistochemistry after exposing the fish to EE. The results revealed that zebrafish exposed to VPA exhibited social deficits, reduced cerebellar cortex volume, and a decrease in c-Fos-positive cells in the Purkinje layer. In contrast, VPA-exposed fish treated with EE showed increased socialization, augmented cerebellar cortex volume, and an elevation in c-Fos-positive Purkinje cells. These findings suggest that alterations induced by VPA may be ameliorated through EE treatment, highlighting the potential therapeutic impact of sensory stimulation in conditions related to ASD.

**Keywords:** ASD; cerebellum; zebrafish; social behavior; valproic acid; animal model

## 1. Introduction

Autism spectrum disorder (ASD) is a human neurodevelopmental disorder characterized by impairments in social domain, motor alterations, and behavioral inflexibility, commonly accompanied by other comorbidities [1–3]. ASD has both genetic and non-genetic etiologies, which act as risk factors in autism development [4,5]. One such high-risk factor is exposure to teratogenic agents during embryogenesis, exemplified by valproic acid (VPA) [6]. This anticonvulsant drug is widely employed as an inducer of ASD in animal models across various species [7–10]. Consistent with findings in autistic humans, exposure to VPA during embryogenesis in zebrafish, is associated with behavioral, neuroanatomical, and genetic alterations, along with deficits in neuronal connectivity and neurotransmitter expression. Multiple studies have shown that VPA acts as a histone deacetylase inhibitor, which partly explains its teratogenic effects [11–15]. The zebrafish model of ASD offers methodological advantages, including high socialization within the species, prolific reproductive performance, a broad behavioral repertoire, physiological and genetic homology, and neuroanatomical similarity to mammalian species, including humans [13].

While central nervous system (CNS) alterations in individuals with autism are highly heterogeneous, it is acknowledged that the cerebellum is one of the structures exhibiting significant abnormalities. Specifically, there is observed cortical volume reduction, diminished number, and size of Purkinje neurons (PN), as well as disruptions in their connectivity

and neurotransmitter expression [16–19]. Traditionally associated with motor and learning functions, the cerebellum's involvement in cognitive, social, and emotional functions was demonstrated recently [20–24]. Consequently, alterations in cerebellar connectivity and morphology are now associated with behavioral deficits in motor, cognitive, and affective domains [16,18,25].

In this regard, the intracerebellar and cerebrocerebellar connectivity in zebrafish is akin to that observed in other vertebrates. Cerebellar information is predominantly processed by Purkinje neurons (PN), which receive direct or indirect input from intracerebellar interneurons and eurydendroid cells [26]. The PNs also exhibit interconnections with other PNs through axonal collaterals, facilitating the synchronization of the cerebellar network [27]. Primary input to PNs is received from granule and stellate cells, while their output is directed towards efferent cells, primarily in the telencephalon, optic tectum, and thalamic nuclei [28,29]. Most projections from the medial region of the cerebellar crest relay through thalamic nuclei before reaching the telencephalon, suggesting that this area plays a role in cognitive and socio-affective information processing. Additionally, the cerebellum is an evolutionarily conserved structure among vertebrates, including zebrafish, allowing for comparisons across species due to its shared cytoarchitecture [27,28,30].

Regarding therapeutic approaches in ASD, a considerable portion of pharmacological treatments is employed to address various comorbidities. However, the treatment of social deficits is predominantly non-pharmacological. One such strategy is sensory stimulation, where improvement in certain social and adaptive skills in individuals with autism has been demonstrated [31,32]. Similarly, in laboratory settings, sensory stimulation is utilized as a treatment in ASD animal models through enriched environments (EE). In this context, EE involves exposing individuals to social, structural, dietary, and cognitive stimuli, promoting complex behaviors, and enhancing their quality of life [33,34]. This type of treatment has been shown to improve key behavioral aspects in various animal species, including zebrafish. However, the impact on the neuroanatomical and molecular levels in the cerebellum requires further experimental evidence [32,35–37].

In this context, our laboratory employed zebrafish to assess the effects of EE on the cerebellum of fish exposed to VPA. The initial approach focused on behavioral aspects, specifically preference and social affiliation in adult zebrafish. This was done to evaluate the relevance of the model and the impact of EE on socialization. At the anatomical level, stereology was utilized to calculate the volume of the molecular layer of the cerebellum, aiming to explore phenotypes consistent with an ASD model and the potential effects of EE on cerebellar morphology. Finally, at the molecular level, immunodetection was employed to assess the expression of the c-Fos protein in the Purkinje layer following exposure to social stimuli, as well as the Hud/Huc protein complex to estimate neuronal populations in the Purkinje layer. This was conducted with the intention of exploring the effects of VPA on neuronal activity and population, as well as the potential impact of EE on these parameters.

## 2. Results

### 2.1. EE Improves Socialization in VPA-Treated Zebrafish

The social preference test allows the assessment of the zebrafish's natural tendency to spend more time near conspecifics. The results of the statistical analysis indicated a significant decrease in the total socialization time in zebrafish exposed to VPA compared to typically developing ones. On the other hand the group, Enriched + VPA fishes, showed a significant increase in social preference compared to the VPA group, as well as the control groups. There was a significant effect of EE ($F = 16.38$, DFn = 1, DFd = 20, $p = 0.0006$), contributing 38.81% to the observed total variation. Additionally, a significant interaction between EE and VPA treatment ($F = 5.81$, DFn = 1, DFd = 20, $p = 0.0257$) was identified, contributing 13.76% to the total variability (Figure 1C).

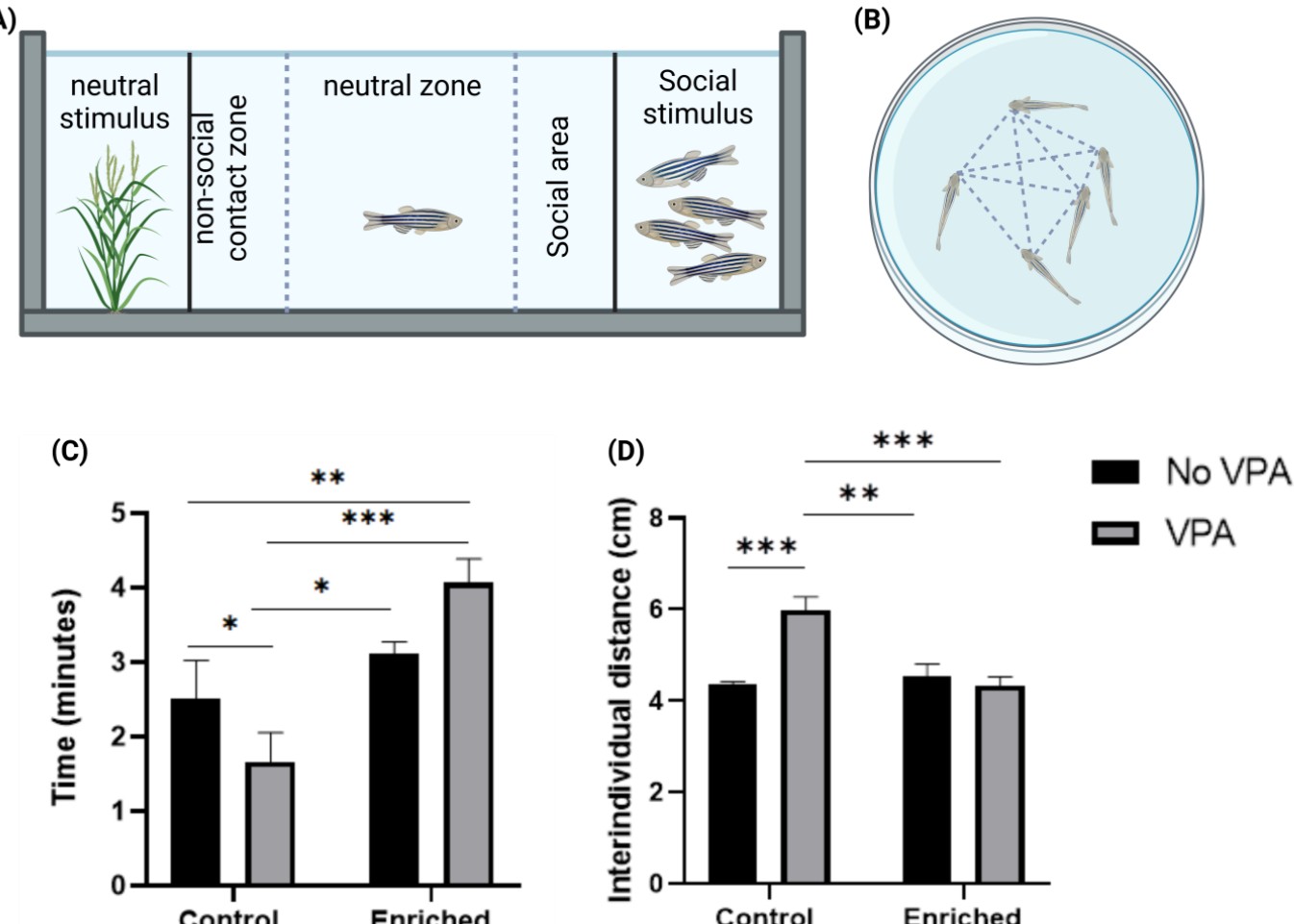

**Figure 1.** (**A**) Representation of social preference apparatus. (**B**) Representation of social affiliation apparatus. (**C**) Socialization test. The graph displays the results of the social preference test for the four experimental groups (n = 8 animals per group). Data show the mean ± SEM. * = $p < 0.05$. ** $p = < 0.005$. *** = $p < 0.0005$. (**D**) Social affiliation test. The graph displays the results of the social affiliation test for the four experimental groups (n = 4 shoals per group). Data show the mean ± SEM. ** $p = < 0.005$. *** $p = < 0.0005$.

The social affiliation test allows the evaluation of the interindividual proximity among the fish in a shoal. The results of this assessment and the statistical analysis indicate that VPA-treated fish have a higher average interindividual distance compared to the control group. On the other hand, Enriched + VPA fishes maintained a distance like the control. The comparison between control groups did not show significant changes due to the EE treatment. A significant interaction of EE and VPA treatment was identified (F = 17.17, DFn = 1, DFd = 12, $p = 0.0014$), contributing 33.2% to the total variance. VPA treatment had a 21.06% impact on the total variability (F = 10.89, DFn = 1, DFd = 12, $p = 0.0063$), while EE treatment contributed 22.55% (F = 11.66, DFn = 1, DFd = 12, $p = 0.0051$) (Figure 1D).

### 2.2. The Volume of the Molecular Layer in VPA-Treated Fish Exposed to EE Is like the Control Fish

Stereological analysis allows the estimation of the volume of structures through mathematical reconstruction. The results of the statistical procedure indicate a significant reduction in the volume of the molecular layer (ML) of the cerebellar crest in the VPA group compared to the control group. The results show that VPA treatment has a significant effect (F = 7.90, DFn = 1, DFd = 20, $p = 0.0108$), representing 21.76% of the total variance. Additionally, there is an effect close to statistical significance for EE treatment (F = 4.14, DFn = 1, DFd = 20, $p = 0.0552$), contributing 11.42% to the total variance. The interaction

between both treatments shows a trend toward significance (F = 4.25, DFn = 1, DFd = 20, $p = 0.0525$), representing 11.71% of the total variance (Figure 2).

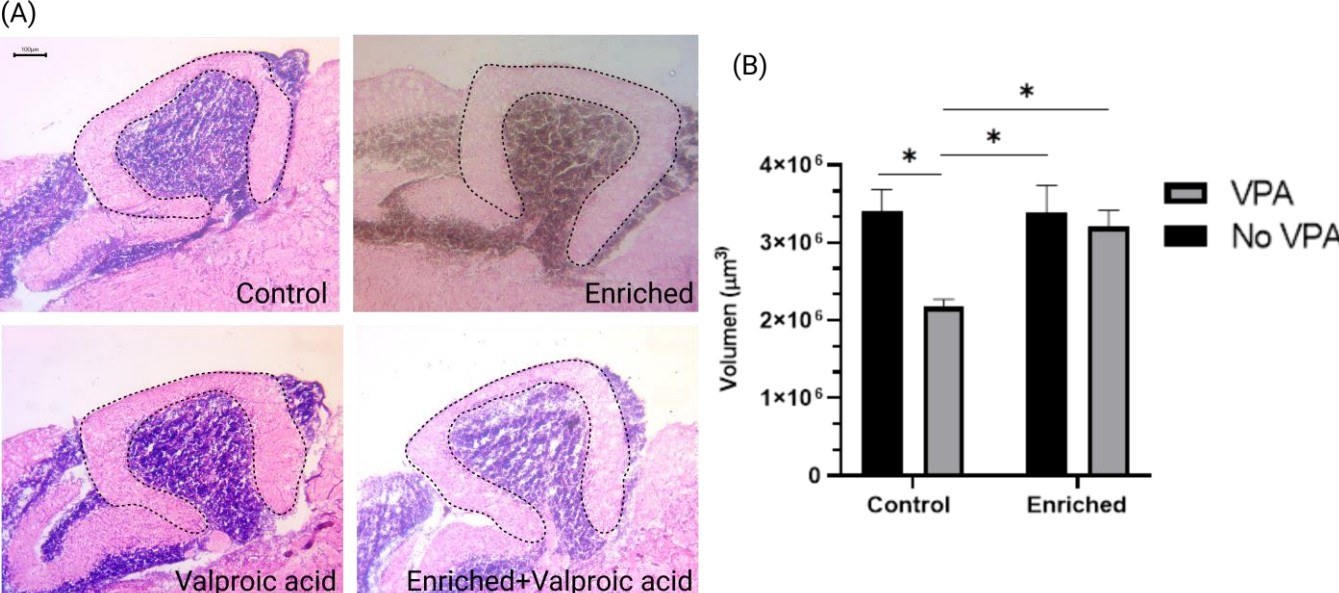

**Figure 2.** (**A**) Sagittal view of the cerebellum stained with hematoxylin and eosin in the four experimental groups. The dashed line marks the boundaries of the molecular layer (ML) of the cerebellar crest (CCe), as considered for stereological evaluation. (**B**) Volume of ML in cerebellar crest. The graph presents the comparison of the volume of the molecular layer (ML) in different experimental groups (n = 8 animals per group). Data show the mean ± SEM. * = $p < 0.05$. Calibration bar at 100 microns.

*2.3. EE Increases the Number of c-Fos Positive Cells in VPA-Treated Fish Exposed to EE*

Estimations of the c-Fos immunoreactive cell population in the CCe were performed. The results revealed a significant decrease in the number of c-Fos immunoreactive cells in animals treated with VPA without EE treatment. However, this effect was not observed in VPA-treated animals exposed to EE, where the expression of c-Fos was similar to control animals. The results show a significantly high effect of VPA treatment (F = 26.07, DFn = 1, DFd = 16, $p = 0.0001$), contributing 48.25% to the total variance. Similarly, there is a significant effect of EE treatment (F = 8.66, DFn = 1, DFd = 16, $p = 0.0096$), representing 16.02% of the total variance. Regarding the interaction between both treatments, although not reaching definitive statistical significance (F = 3.30, DFn = 1, DFd = 16, $p = 0.0879$), it shows a trend towards importance, contributing 6.113% to the total variance. (Figure 3B).

*2.4. Neuronal Counting of the NP*

To explore the effects of EE on the neuronal population of VPA-treated zebrafish, the neuronal population of the NP of the CCe was estimated through immunohistochemistry for Hud/Huc. The results of cell counting and descriptive statistics allow us to appreciate some changes associated with the treatments. On the one hand, VPA-treated fish show a trend towards a reduced number of neurons in the NP of the CCe, while this trend is not observed in VPA-treated fish exposed to EE. Although the number of individuals analyzed in this category is not sufficient to establish significant differences, the description of the statistical trend points to the expected results.

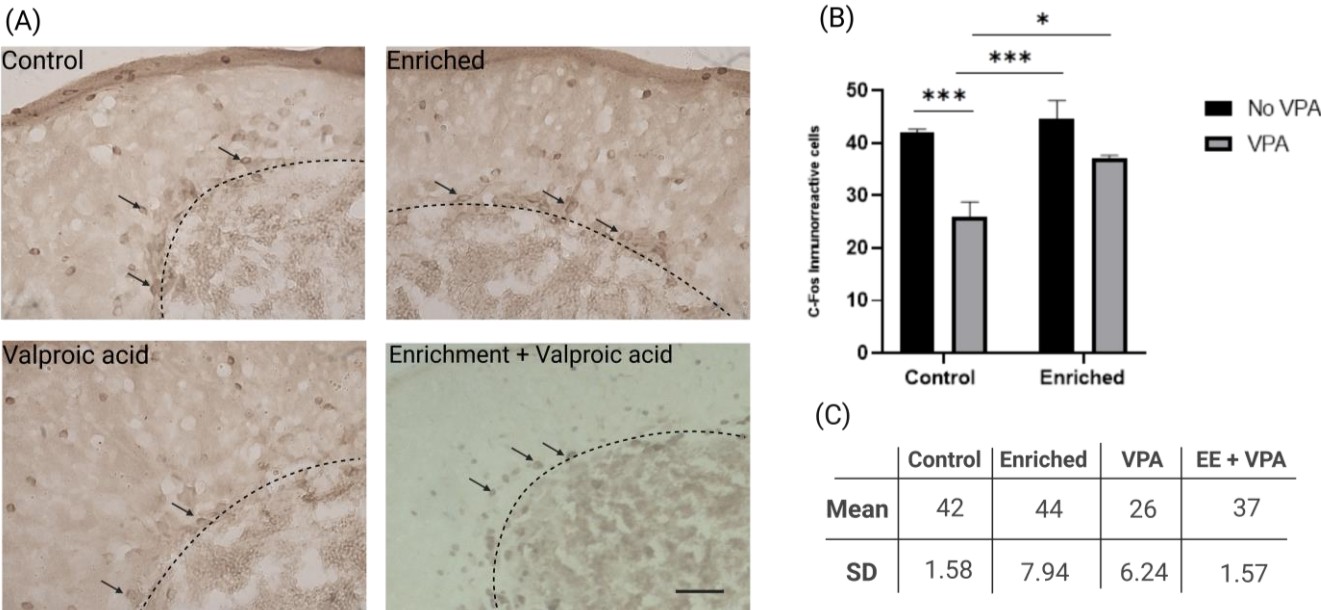

**Figure 3.** (**A**) Immunohistochemistry photomicrographs for c-Fos in the apical zone of the cerebellum in a sagittal section in the four experimental groups. Black arrows indicate c-Fos immunoreactive cells in the Purkinje layer. The dashed line indicates the boundary between the molecular layer and the granular layer. Calibration bar at 100 microns. (**B**) Graph 4: Number of c-Fos immunoreactive cells. The graph presents the comparison of cell counting results between experimental groups (n = 5 animals per group). Data show the mean ± SEM. * = $p < 0.05$. *** = $p < 0.0005$. (**C**) Table of neuronal count for c-Fos positive cells.

### 3. Discussion

The results of the statistical analysis indicate that embryonic exposure to VPA induced significant deficits in social preference and affiliation behavior, a decrease in the volume of the cerebellar CM, as well as a reduction in the number of c-Fos and Hud/Huc immunoreactive cells. Conversely, treatment with EE significantly improved socialization behaviors and was associated with a CM volume and the number of c-Fos and Hud/Huc-positive cells similar to the control groups.

Exposure to VPA from embryonic day 0 to 48 h post-fertilization at a concentration of 48 μM led to a decrease in socialization time and an increase in interindividual distance in zebrafish, indicative of social deficits. This aligns with findings in other studies reporting similar social domain deficits following exposure to VPA at concentrations of 5, 50, and 500 μM during embryonic stages [38]. Additionally, behavioral deficits were observed in response to VPA exposure during later developmental periods (4–5 dpf), suggesting that there may be other vulnerable periods to teratogenic agents during zebrafish neurodevelopment. This vulnerability is likely attributed to the underdeveloped social circuitry during the early stages, as zebrafish exhibit social behaviors within the first week of life, with consolidation occurring by the third week [39]. Thus, insults disrupting the development of this circuitry may result in behavioral alterations [40–42].

In this study, exposure to EE significantly improved socialization behaviors in VPA-treated zebrafish. These findings are consistent with previous research in rodents and humans with autism, where environmental enrichment was shown to enhance sociability and other behavioral parameters [43,44]. The positive effects of EE in the zebrafish VPA model, including improvements in sociability, reproductive behaviors, and anxiety-related behaviors, are in line with previous reports [37,44]. Notably, EE impact on the oxytocinergic system, a crucial regulator of prosocial behaviors, may contribute to these improvements, highlighting a potential avenue for future investigations [45–47].

This study also explored the effects of VPA and EE on the volume of the cerebellum ML using stereology, revealing a significant reduction in CM volume in the VPA-exposed group compared to the typical development group. This aligns with reports of cerebellar hypoplasia in both imaging studies and postmortem tissues from individuals diagnosed with ASD [16,48,49]. The effects of VPA on cerebellar development were further supported by investigations into Wnt-catenin signaling, a pathway crucial for cerebellar development, which demonstrated increased expression in VPA-exposed zebrafish [39,50].

Neurogenesis in the zebrafish cerebellum initiates around 16 h post-fertilization (hpf) with the establishment of the "cerebellar territory", and histogenesis begins around 48 hpf [27,28]. This coincides with the period of VPA exposure in this study, suggesting that the drug has significant effects on cellular organization during cerebellar histogenesis. The observed behavioral and morphological phenotypes align with characteristics associated with ASD, such as reduced brain size and dysregulation of neuronal proliferation and migration [51–55]. Previous studies delving into the deregulation of neuronal proliferation due to VPA exposure in embryo zebrafish demonstrated the impact on the upregulation of the Notch pathway, which is also involved in cerebellar histogenesis [12]. They also showed that the impact of VPA varies depending on the embryonic exposure period. Embryo exposure resulted in decreased neuronal proliferation, whereas adult exposure led to increased proliferation.

Notably, zebrafish exposed to both VPA and EE did not show a reduction in CM volume, suggesting a protective effect of EE on cerebellar morphogenesis in VPA-exposed fish. The potential neuroprotective and compensatory effects of EE on the cerebellum, demonstrated by increased expression of brain-derived neurotrophic factor (BDNF) and glutamic acid decarboxylase (GAD65/67) in rodent models, may extend to zebrafish [53,56,57]. These proteins play essential roles in neurodevelopment, particularly in maintaining a balance between neuronal inhibition and excitation in the cerebellum.

The analysis of c-Fos expression in the Purkinje cell layer (CP) of the cerebellum following social stimulation in a social preference paradigm provided insights into cerebellar involvement in behavior. VPA exposure was associated with reduced c-Fos expression in the CP, potentially indicating alterations in cerebellar connectivity. This could be attributed to changes in genes like Shank3, known for its involvement in synapse formation and frequently altered in individuals with ASD and in response to VPA exposure [58–60]. Regarding this, Shank3 is a protein that plays a major role in the formation and maintenance of synapses, especially in the cerebellum, and is often found to be altered in individuals with ASD and in the VPA zebrafish model [39,49–60]. Consequently, it has been proposed as one of the genetic alterations of significant importance in ASD [25].

Another hypothesis for the decreased c-Fos expression could be a reduction in Purkinje cell number, significantly associated with VPA exposure [60]. While a neurotypical cerebellum has each Purkinje cell receiving input from a single climbing fiber, an ASD individual may receive input from approximately four climbing fibers per Purkinje cell, suggesting altered intracerebral circuit connectivity [25].

In contrast, zebrafish exposed to both VPA and EE exhibited c-Fos expression in the CP comparable to control groups, suggesting positive effects of EE on cerebellar circuit organization. Rodent studies have reported protective and reorganizational effects on the Purkinje cell layer in response to EE, including increased dendritic tree size and the number of mature synapses in the cerebellum [61]. This protective effect may be associated with increased neuroplasticity, induced by constant sensory stimulation, contributing to changes in cellular morphology and synaptic populations [61–65].

The effect of EE on ASD-like phenotypes has also been reported in murine models exposed to VPA during embryogenesis. Improvements were reported in socialization and the expression of neurotransmitters at the central and peripheral level, which are consistent with what was observed in this work [63]. However, the VPA zebrafish model stands out for its easy handling, high socialization, and low maintenance cost. Furthermore, due to its development ex utero, it facilitates exposure to teratogenic substances during critical stages

of neurodevelopment; in contrast to murine models in which VPA is usually administered intraperitoneally. Zebrafish also display robust social behavior, often living in large shoals of conspecifics, which facilitates the study of social behavior and any alterations in this domain. Therefore, the embryonic exposure model to VPA in zebrafish is emerging as an important model to study possible pharmacological and non-pharmacological treatments for ASD [65].

Regarding possible cellular and molecular mechanisms of EE on neuronal organization and activity, the most accepted hypothesis is that EE increases neuroplasticity in the central nervous system [36,61–65]. Neuroplastic changes occur in response to constant stimuli activating neural pathways concurrently. This process, known as synaptic plasticity, involves the formation of new synapses and the reorganization of existing ones to optimize connections between different brain areas [32,35,65]. Early and consistent sensory stimulation by EE is hypothesized to modify individual cells and the overall cerebral circuitry, potentially explaining the reported results.

The results highlight the teratogenic effects of VPA in inducing ASD-like characteristics in zebrafish, affecting behavior, morphology, and cellular activity. Importantly, these alterations were mitigated by exposure to EE, indicating positive effects of sensory stimulation in neurodevelopmental disorder models.

## 4. Materials and Methods

### 4.1. Subjects

Adult wild-type zebrafish (*Danio rerio*) were utilized for the study, sourced from the aquarium at the Institute of Brain Research of the Universidad Veracruzana (IICE) and housed in 20-L tanks under standard laboratory conditions [66–68]. All procedures were conducted in accordance with the Mexican standard NOM-062-ZOO-1999 for animal care and handling [66]. Experimental and manipulative procedures received approval from the Institutional Committee for the Care and Use of Laboratory Animals (CICUAL) at the IICE.

### 4.2. Embryo Collection

To obtain fertilized embryos, a male and a female with typical development were housed in a 3.5-L breeding tank in the afternoon (4:00 p.m.). Fertilized eggs were collected at 12:00 p.m. the following day, following previously established protocols [69,70]. Subsequently, the fertilized embryos were transferred to 6-well cell culture plates, evenly distributed for pharmacological exposure. In each well, with a capacity of 12 mL, 15 fertilized embryos were placed.

### 4.3. Pharmacological Treatment

Exposure to valproic acid (Sigma-Aldrich, Naucalpan, Mexico) was carried out following standardized protocols [15], administered by immersion at a concentration of 48 μM, and prepared with aquarium system water from 0 to 48 hpf. At 48 hpf, any remaining VPA residues were carefully removed by washing the embryos. To maintain experimental conditions, the control group underwent the same procedure using clean aquarium system water.

### 4.4. Environmental Treatments

After hatching (72 hpf), control and VPA-treated larvae were evenly divided into two groups and housed in two environmental configurations: enriched environment and standard environment. In this way, 4 experimental groups with different conditions were formed (Figure 4A).

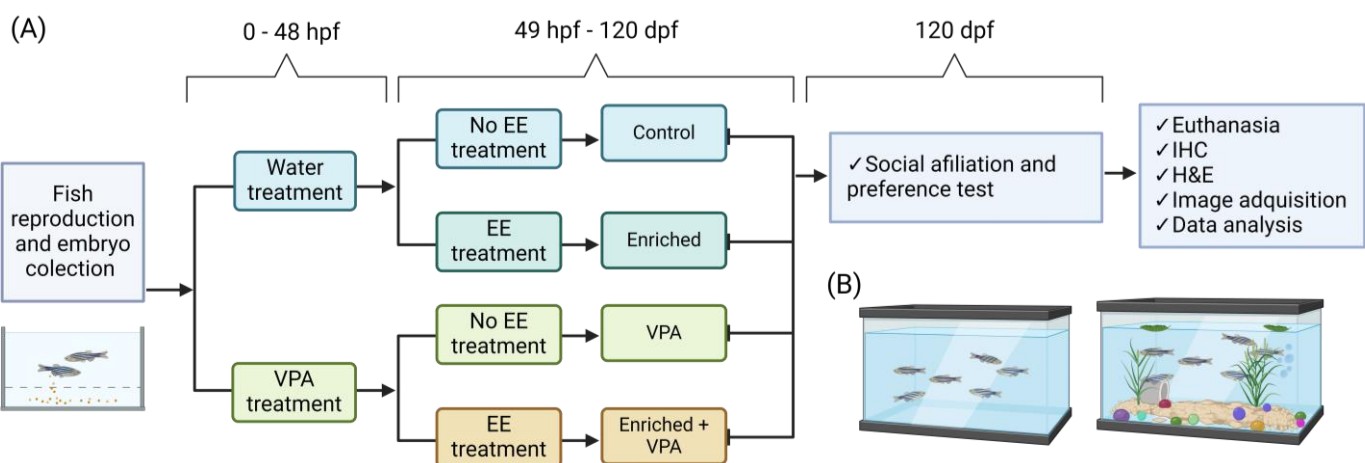

**Figure 4.** (**A**) Representative diagram of the experimental design. Embryos resulting from animal reproduction are subjected to either VPA or tank water as a control for the initial 48 h of development. Subsequently, the groups are stratified based on environmental enrichment treatment or standard housing, serving as control. Behavioral assessments are conducted at 120 dpf, followed by euthanasia for brain extraction and subsequent IHC and H&E analysis. (**B**) Representation of fish tanks with enriched environment treatment and without enriched environment.

Control Group: This group was not exposed to any additional treatment, serving as a standard reference.

Enriched Environment: This group of zebrafish was exclusively exposed to an enriched environment.

Valproic Acid: This group was exclusively exposed to valproic acid.

Enriched Environment + Valproic Acid: This group was exposed to both the enriched environment and valproic acid.

For the EE, transparent 3.5-L tanks were used, containing elements such as colorful substrate stones, plastic plants, colored stones and marbles, plastic tubes, and bubble devices. Feeding in the EE was supplemented twice a week with microalgae. These elements were modified weekly to prevent habituation and occupied one-quarter of the tank. In the case of the CE, 3.5-L tanks without structural additions were used, covered with a semi-transparent film. All fish were maintained in these treatments until 120 dpf at a population density of 3 zf/L [71] (Figure 4B).

### 4.5. Behavioral Assessment

The tests were conducted in the age range of 110 to 120 dpf using two social domain behavioral tests built in our laboratory. Both tests took place between 10:00 am and 02:00 pm, with a five-minute habituation period. The tests were recorded from an overhead view using an iPhone 5 (Apple®, Cupertino, CA, USA), and the videos were analyzed using the RealFishTracker software (version 0.4.0).

#### 4.5.1. Social Preference Test

Social preference was evaluated in a glass test tank measuring 40 cm × 10 cm × 12 cm, with two transparent dividers forming three compartments (two lateral and one central). In one of the lateral compartments, a plastic plant was placed as a neutral stimulus, and in the other, 5 zebrafish of typical development were positioned as a social stimulus, matching in size and age to the test fish. The test fish was placed in the central space, and the time spent near conspecifics was evaluated over a period of 6 min [69–72] (Figure 2A).

#### 4.5.2. Social Affiliation Test

The social affiliation test involved placing a school of 5 fish from the same experimental group in a cylindrical tank measuring 14.5 cm in diameter with a capacity of 1 L.

Six minutes of free activity were recorded, and the interindividual distance between all fish was measured and divided by the number of individuals, to calculate a social affiliation index (Figure 2B) [72].

### 4.6. Brain Extraction

After a 90-min interval following the social preference test, the fish were euthanized using the reduction of temperature and decapitation method at the dorsal fin level [73,74]. The head was fixed in 4% paraformaldehyde (PFA) for 30 min, and complete brains were extracted, visualizing the structure with a dissecting microscope (Meiji EMZ-TR, Meiji Techno Co., Saitama, Japan). Subsequently, the brains were kept in 4% PFA for 24 h and then in 30% sucrose until processing. The whole brains were sectioned in a sagittal plane using a cryostat (Leica, CM1850, Wetzlar, Alemania) at a thickness of 20 μm and at −24 °C. The sections were mounted on gelatin-coated slides and stored at 4 °C until further processing.

### 4.7. Histology

For histological processing, 8 animals per group were used, and 5 cryosections were selected for each fish. The samples were processed for hematoxylin and eosin (HE) staining and mounted with Permount (Fisher Chemical SP15-500, Madrid, España) following our laboratory's established protocol. Subsequently, images of the entire cerebellum were acquired using an AX70 OLYMPUS microscope at 10× magnification.

### 4.8. Image Analysis and Volumetric Estimation

To calculate the volume of the ML of the cerebellum, H&E-stained sections were used. Using ImageJ (FIJI) software (version 1.54f), the area of ML was calculated, considering the limit of the Purkinje cell layer and the molecular layer limit in 5 nonconsecutive cerebellar sections (n) at the midline level. The formula for calculating the volume involved measuring the area (A), then multiplying it by the thickness of the sections (20 μm) (T), and multiplying by the distance between (d) the sections (100 μm) [v = ΣA·T·d·n]. A digitized atlas of the adult zebrafish brain served as a reference to define the lobes' limits and their relative anatomical locations, facilitating accurate identification and measurement of the ML volume [75].

### 4.9. Immunohistochemistry

Immunohistochemical detection was performed following previously established protocols [76,77]. Five animals per group were used, and 3 non-consecutive sections (separated by 100 μm) were selected for immunodetection. Cryosections mounted on gelatin-coated slides were washed for 5 min with 0.1 M PBS and 0.1% PBS-T, followed by washing with 0.1 M phosphate buffer (PBS) and 30% $H_2O_2$ to eliminate endogenous peroxidases. Preblocking was carried out with a 30-min incubation in 0.3% BSA PBS-T at a concentration of 3:100. Subsequently, the samples were incubated with the primary antibody for c-Fos (c-Fos [E-8] sc-166940, Santa Cruz, mouse monoclonal IgG1) or Huc-Hud (Anti-Huc-Hud abcam—rabbit polyclonal), diluted in the preblocking solution at a concentration of 1:500 for 24 h at 4 °C. Afterwards, two washes with 0.1 M PBS and one wash with 0.1% PBS-T were performed, followed by a 2-h incubation with the secondary antibody (Biotin-SP-conjugates AffiniPure Goat Anti-mouse IgG) diluted in 0.3% PBS-T at a concentration of 1:100. The samples were incubated for 90 min in a solution of 0.1% PBS-T with 1 drop of Biotin and 1 drop of Avidin (ABC complex) and revealed with DAB (VectorDAB peroxidase substrate SK-4100) at a concentration of 3:10. The same procedures were performed for positive controls, excluding the incubation with the primary antibody.

### 4.10. Cell Counting

Images were obtained with an AX70 OLYMPUS light microscope at 10× and 20× magnifications of the cerebellar crest lobe. The images were processed with ImageJ (FiJi, version, 1.54f), and cells positive for c-Fos and Hud/Huc in the Purkinje layer were

counted. A digitized atlas of the adult zebrafish brain was used as a reference for defining lobe boundaries and relative anatomical location [75].

### 4.11. Statistical Analysis

The results of total time and interindividual distance in behavioral tests, as well as cerebellar volume and cell counting, are presented as $\pm$ SEM. Data were statistically analyzed using a two-way ANOVA with Prism 9.5 software (GraphPad, San Diego, CA, USA) and Tukey's post hoc test. Statistical significance was considered at $p < 0.05$.

## 5. Conclusions

This study provides insight to the effects of environmental enrichment (EE) on the cerebellum of zebrafish treated with valproic acid (VPA) during embryogenesis. In summary, the conclusions drawn from this research are as follows:

(a) Embryonic exposure to VPA in zebrafish is associated with reduced socialization time and an increase in interindividual distance. Prolonged and consistent exposure to EE significantly improves social preference and affiliation behaviors in VPA-treated zebrafish. (b) VPA treatment is significantly associated with a decreased volume of the cerebellar CM in zebrafish. Exposure to EE significantly prevents the reduction in CM volume associated with VPA. (c) VPA reduces the number of c-Fos immunoreactive cells in the cerebellar CP of zebrafish. EE exposure prevents the effect of VPA on c-Fos expression in the CP. (d) VPA appears to reduce the number of neurons in the CP. EE exposure seems to induce a preliminary increase in the neuronal population in VPA-treated fish.

Collectively, the findings of this study underscore the importance of considering the impact of substance exposure during embryonic development in the etiology of autism spectrum disorder (ASD) in zebrafish. The study suggests that EE has therapeutic potential in improving symptoms related to sociability and cerebellar function in animal models.

**Author Contributions:** Conceptualization, B.F.-P. and M.R.T.-C.; methodology, B.F.-P. and M.R.T.-C.; validation, B.F.-P., F.C.-S., J.M., M.E.H.-A., A.G.C.-A., D.H.-C., F.R.-D., G.E.A.-A., C.A.P.-E. and M.R.T.-C.; formal analysis, B.F.-P., J.M., A.G.C.-A. and M.R.T.-C.; investigation, B.F.-P. and M.R.T.-C.; resources, B.F.-P. and M.R.T.-C.; data curation, B.F.-P., F.C.-S., J.M., M.E.H.-A., A.G.C.-A., D.H.-C., F.R.-D., G.E.A.-A., C.A.P.-E. and M.R.T.-C.; writing—original draft preparation, B.F.-P. and M.R.T.-C.; writing—review and editing, B.F.-P., F.C.-S., J.M., M.E.H.-A., A.G.C.-A., D.H.-C., F.R.-D., G.E.A.-A., C.A.P.-E. and M.R.T.-C.; visualization, B.F.-P., F.C.-S., J.M., M.E.H.-A., A.G.C.-A., D.H.-C., F.R.-D., G.E.A.-A., C.A.P.-E. and M.R.T.-C.; supervision, M.R.T.-C.; project administration, M.E.H.-A. and M.R.T.-C.; funding acquisition, B.F.-P. and M.R.T.-C. All authors have read and agreed to the published version of the manuscript.

**Funding:** This research was funded by Conahcyt Fellowship 1101287 to B.F.-P.

**Institutional Review Board Statement:** The animal study protocol was approved by the Ethics Committee (CICUAL) of IICE (Instituto de Investigaciones Cerebrales). Protocol code 2021-012a and date of approval: 20 April 2022).

**Informed Consent Statement:** Not applicable.

**Data Availability Statement:** The data supporting the findings of this study are available upon reasonable request. For additional information or specific protocols, please feel free to contact the corresponding author at bersebaflo@hotmail.com.

**Acknowledgments:** The authors would like to express their gratitude to Lizbeth Donaji Chi-Castañeda for her indispensable support throughout this research. Special thanks are also extended to Rey-Murrieta for his generosity and to María de la Paz Palacios Arellano, Mariana Aguirre Rebolledo for their indispensable support. Additionally, the authors appreciate the support and collaboration of their colleagues at the IICE.

**Conflicts of Interest:** The authors declare no conflicts of interest.

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
