# Peer review of "Effect of Enriched Environment on Cerebellum and Social Behavior of Valproic Zebrafish"

_neurosci, doi:10.3390/neurosci5020009_

Round 1
Reviewer 1 Report
Comments and Suggestions for Authors
This manuscript studies the effects of exposure to valproic acid (VPA) on zebrafish and the potential reversal of these effects by enriched environment (EE). The researchers used zebrafish as a model organism because they are genetically similar to humans and their brains develop rapidly.
Authors found that zebrafish exposed to VPA during embryonic development exhibited social deficits, also had a smaller cerebellum, a brain region involved in movement, learning, and social interaction. Additionally, these fish had fewer Purkinje neurons, which are important for cerebellar function. Is it possible for the authors to manually count the no. of c-Fos immunoreactive cells in each group and represent it in graphical form? Authors couldn’t get required number of cells to quantify the data of neuronal counting. It would`ve been a significant addition to the story.
These findings may suggest that VPA can cause ASD-like symptoms in zebrafish, and that enriched environment may be a helpful treatment strategy. Authors suggest that EE may work by increasing neuroplasticity, which is the brain's ability to change and adapt. We agree on that too.
This study is important, thought not the first, because it provides a better understanding of how ASD may develop and how environmental factors may influence its symptoms. It also suggests that EE may be a potential treatment for ASD in humans.
Author Response
- The overall presentation of the results has been enhanced, as suggested by the reviewer, particularly in the sections concerning figures and the drafting of their captions.
- A summary table containing cellular counting data for c-Fos was created and included in figure 3, in accordance with the reviewer's suggestion.
- With regards to the cellular counting of Hud/HUC, it was not originally part of the study's objectives. However, we believe it could enhance the understanding of the data as a preliminary approach, as it enables the evaluation of specific neuronal populations. We intend to conduct further investigations involving this protein in future studies.
On behalf of the authors, we appreciate the suggestions and comments provided by the reviewers to enhance the quality of our manuscript.
Reviewer 2 Report
Comments and Suggestions for Authors
In this manuscript, authors showed an improved social behavior in enriched environment conditions in valproic acid-induced ASD in the Zebrafish model. Authors used combination of methods such as behavior tests and immnuohistochemistry.
I have a few concerns which are below in point wise.
1. The overall presentation of the data is very poor. Authors need to re-organize the data. For example, image and corresponding quantification graph should be in one figure. Additionally, image would come first and the the quantification graph.
2. Authors also need to change the format of referencing, such as reference number should be in superscript format.
3. In line 93, "There was a significant effect of EE (f = 16.38, p = 0.0006), contributing 38.81% to the ob-93 served total variation" Here, f=16.38 is not described and the value does not appear in the graph.
4. In the cFos data, author shown in graph that c-Fos number is decreased in the valproic-treated group. However, the representative image show there are more c-Fos numbers.
Author Response
- The general presentation of the data, as well as the graphs, has been improved, as recommended by the reviewer.
- The reference formatting has been corrected.
- Detailed statistical data have been added for all results, as requested by the reviewer.
- The representative image of cellular counting for c-Fos has been corrected, as recommended by the reviewer.
On behalf of the authors, we appreciate the suggestions and comments provided by the reviewers to enhance the quality of our manuscript.
Reviewer 3 Report
Comments and Suggestions for Authors
After reading the manuscript my major concerns are as follows:
- Add information on a post-hoc test used to analyze the data with two-way ANOVA. It is crucial for illustrating significance on graphs 1-4.
- Please discuss the problem that VPA is a histone deacetylase inhibitor and it significantly affects proliferation in zebrafish. This is critically important in light to the fact that zebrafish were exposed to VPA from 0-48 hpf. (see: Dozawa M, Kono H, Sato Y, Ito Y, Tanaka H, Ohshima T. Valproic acid, a histone deacetylase inhibitor, regulates cell proliferation in the adult zebrafish optic tectum. Dev Dyn. 2014 Nov;243(11):1401-15. doi: 10.1002/dvdy.24173. ; He Y, Cai C, Tang D, Sun S, Li H. Effect of histone deacetylase inhibitors trichostatin A and valproic acid on hair cell regeneration in zebrafish lateral line neuromasts. Front Cell Neurosci. 2014 Nov 13;8:382. doi: 10.3389/fncel.2014.00382.) .
- Provide information about superiority of the zebrafish model over the mouse model prenatally exposed to VPA. This is the main rationale of this study.
- Please, discuss the fact that VPA affects Autism Spectrum Disorders in mice prenatally exposed to VPA (see: Zappala C, Barrios CD, Depino AM. Social deficits in mice prenatally exposed to valproic acid are intergenerationally inherited and rescued by social enrichment. Neurotoxicology. 2023 Jul;97:89-100. doi: 10.1016/j.neuro.2023.05.009., Zahedi E, Sadr SS, Sanaeierad A, Roghani M. Valproate-induced murine autism spectrum disorder is associated with dysfunction of amygdala parvalbumin interneurons and downregulation of AMPK/SIRT1/PGC1α signaling. Metab Brain Dis. 2023 Aug;38(6):2093-2103. doi: 10.1007/s11011-023-01227-1.)
- What differences in social functioning were observed between zebrafish and mice when both exposed to VPA. Please, discuss the problem more deeply, not superficially.
- What cellular pathways were activated in zebrafish by VPA in relation to Autism Spectrum Disorders? Were the same as in mice or quite different?
- What were the main differences between zebrafish and mice exposed prenatally to VPA when testing Autism Spectrum Disorder conditions? (see: Mitsuhashi T, Hattori S, Fujimura K, Shibata S, Miyakawa T, Takahashi T. In utero Exposure to Valproic Acid throughout Pregnancy Causes Phenotypes of Autism in Offspring Mice. Dev Neurosci. 2023;45(5):223-233. doi: 10.1159/000530452.)
No specific comments
Author Response
- Regarding the presentation of post-hoc statistical processing, the methods describe the use of Tukey's multiple comparisons.
- Comments on the effect of AVP on HDAC have been incorporated into the discussion, citing the work of Dozawa et al. (2014), as recommended by the reviewer.
- In our opinion, both the zebrafish VPA model and the murine VPA models possess particularities enabling them to address different species-dependent questions. We briefly discuss some of these differences and similarities, as suggested by the reviewer.
- As recomended by reviewer, a comment on the work of Zappala et al. (2023) has been added to the discussion, briefly describing the relevance of their results when compared to our work. Comparing results across different species enhances our understanding of ASD and its potential treatments.
- From line 238 to 249, the reviewer's suggestions are discussed.
- From line 182 to 208, some of the points suggested by the reviewer are discussed. While the aim of the research and its scope did not involve explaining VPA activation pathways in this model, we deem it important to discuss them based on other studies to better comprehend and interpret our results.
- This observation is included within responses 1, 4 and 6.
On behalf of the authors, we appreciate the suggestions and comments provided by the reviewers to enhance the quality of our manuscript.
Round 2
Reviewer 2 Report
Comments and Suggestions for Authors
The manuscript is improved significantly by the authors.
I don't have any further issues.
Author Response
Thank you for your observations
Reviewer 3 Report
Comments and Suggestions for Authors
line 39 - it should be "model" not "mr4del".
line 553: reference no: 68. Provide full details about the journal name and pagination.
Comments on the Quality of English Language
No comments
Author Response
Thank you for your observations. Changes were made to lines 39 and 553 according to the reviewer's comments